# FREDSum: A Dialogue Summarization Corpus for French Political Debates

**Virgile Rennard[1,2], Guokan Shang[1], Damien Grari[3]**
**Julie Hunter[1], Michalis Vazirgiannis[2]**

[1]Linagora, France [2]École Polytechnique, France [3]Grenoble Ecole de Management, France
`virgile@rennard.org` `guokan.shang@polytechnique.edu`
`damien.grari@grenoble-em.com` `jhunter@linagora.com`
`mvazirg@lix.polytechnique.fr`

## Abstract

Recent advances in deep learning, and especially the invention of encoder-decoder architectures, has significantly improved the performance of abstractive summarization systems. The majority of research has focused on written documents, however, neglecting the problem of multi-party dialogue summarization. In this paper, we present a dataset of French political debates[1] for the purpose of enhancing resources for multi-lingual dialogue summarization. Our dataset consists of manually transcribed and annotated political debates, covering a range of topics and perspectives. We highlight the importance of high quality transcription and annotations for training accurate and effective dialogue summarization models, and emphasize the need for multilingual resources to support dialogue summarization in non-English languages. We also provide baseline experiments using state-of-the-art methods, and encourage further research in this area to advance the field of dialogue summarization. Our dataset will be made publicly available for use by the research community.

## 1 Introduction

Summarization aims at condensing text into a shorter version that optimally covers the main points of a document or conversation. Extant models adopt one of two approaches: while *extractive summarization* aims at creating an optimal subset of the original source document by simply selecting the most important sentences without changing them, *abstractive summarization* aims at reformulating the most important points with novel sentences and words that may not have been seen before in the text. As such, abstractive summarization is a natural language generation task that relies more deeply on the meaning of the source text.

A significant research effort has been dedicated to traditional document summarization (Lin and Ng, 2019), due in part to the fact that high quality data is readily available online, leading to the creation of even multi-lingual datasets for news summarization (Hermann et al., 2015; Kamal Eddine et al., 2021; Evdaimon et al., 2023), and scientific articles (Nikolov et al., 2018).

Such data is much harder to come by for dialogue, although there are a few corpora of multi-party conversation and chat that have been painstakingly created with the aim of furthering research on abstractive meeting and conversation summarization. Some examples include the AMI (Mccowan et al., 2005), ICSI (Janin et al., 2003) and ELITR (Nedoluzhko et al., 2022) corpora for meeting summarization (Shang, 2021) and the SAMSum corpus (Gliwa et al., 2019) for chat summarization (cf. Rennard et al., 2023; Feng et al., 2021, for more discussion.) However, due to the particular nature of dialogical interactions and even different types of dialogue, the relatively small size of conversation data sets compared to those for document summarization and other difficulties posed by spontaneous speech (Rennard et al., 2023), dialogue summarization remains an open task.

Another significant problem for the task of dialogue summarization is the lack of multi-lingual data. While ELITR (Nedoluzhko et al., 2022) contains some meetings in Czech, and VCSum (Wu et al., 2023), meetings in Chinese, there are currently no datasets available for multiparty conversation in other non-English languages. This imposes a significant limitation on the development of solutions for dialogue summarization in multilingual situations. There is thus a significant need for multilingual dialogue summarization resources and research.

In this paper, we present FREDSum, a new dataset for FREnch political Debate Summarization and the first large-scale French multi-party summarization resource. FREDSum is designed to serve as a valuable resource for furthering research

---

[1]https://github.com/VRennard/FreDSum

| | Dataset | Lang. | #Transcripts | #Words/trans. | #Turns/trans. | #Speakers/trans. | #Words/sum. |
|---|---|---|---|---|---|---|---|
| dialog | SAMSum | EN | 16369 | 93.8 | 11.2 | 2.4 | 20.3 |
| | DialogSum | EN | 13460 | 131.0 | - | - | 23.6 |
| | MediaSum | EN | 463596 | 1553.7 | 30.0 | 6.5 | 14.4 |
| meeting | AMI | EN | 137 | 6007.7 | 535.6 | 4.0 | 296.6 |
| | ICSI | EN | 59 | 13317.3 | 819.0 | 6.3 | 488.5 |
| | MeetingBank | EN | 6892 | 3800.3 | 146.9 | 3.2 | 87.2 |
| | VCSum | CN | 239 | 14106.9 | 73.1 | 5.6 | 231.9 |
| | ELITR | EN/CS | 179 | 7549.9 | 884.5 | 6.5 | 327.9 |
| debate | FREDSum | FR | 142 | 2595.5 | 54.2 | 4.2 | 238.9 |
| | FREDSum$_{preS}$ | FR | 1140 | 71386.8 | 685.0 | 53.4 | - |
| | FREDSum$_{preA}$ | FR | 4563 | 28619.8 | 445.8 | 56.0 | - |

Table 1: Comparison between FREDSUM and other news, dialogue or meeting summarization datasets. # stands for the average result.FREDSum$_{preS}$ and FREDSum$_{preA}$ represent the pretraining datasets released along FREDSum.

on dialogue summarization in French and multilingual dialogue summarization more generally. Each debate in the corpus is manually transcribed and annotated with not only abstractive and extractive summaries but also important topics which can be used as training and evaluation data for dialogue summarization models. The debates were selected with an eye to covering a range of political topics such as immigration, healthcare, foreign policy, security and education, and involve speakers with different perspectives, so as to not over-represent a party or an ideology.

We highlight the importance of high quality transcription and annotations for training accurate and effective dialogue summarization models. We also present baseline experiments using state-of-the-art methods and encourage further research in this area to advance the field of dialogue summarization. Our goal is to promote the development of multilingual dialogue summarization techniques that can be applied not only to political debates but also to other domains, enabling a broader audience to engage with important discussions in their native language.

## 2 FREDSum : FREnch Debate Summarization Dataset

Due to the heterogeneous nature of conversations, different dialogue summarization datasets serve different purposes. The uses and challenges inherent to chat data, film or meeting transcripts, Twitter threads, and other kinds of daily communication are heterogeneous. Differences may stem from transcript length, the level of disfluencies, the specific speech style of, say, chatrooms and Twitter threads, the goal oriented manner in which a meet-

ing is conducted, or even the downstream tasks for which a dataset is constructed. The data and the models used to treat can indeed be very diverse.

Debates are multi-party discussions that are generally long and fairly well-behaved both in the cleanness of the language used and their tight organization. Most debates follow the same structure, an initial topic is given, followed by an assertion from a speaker, refutal by others with an alternative, and a further refutal by the original speaker. This systematic organization lends itself well to the study of discourse structure and dialogue dynamics.

In this section, we aim at explaining the elements of annotations provided in FREDSum as well as the instructions followed by the annotators.

### 2.1 Data Collection, and curation

Data collection and curation are critical components of any data driven project. Specifically, in the case of political debates, a dataset needs to cover a wide range of political fields, such as economic policies, social issues and foreign politics, but also needs to be representative of the landscape of the country's politics over the years and thus should not fail to represent any of the major political parties. The debates in FREDSum thus cover a diverse range of topics from 1974 to 2023 of political debates.

Due to the length of political debates, which can go on for mutiple hours and touch on a wide variety of specialized topics, we decided to segment each debate by topic. As such, a presidential debate that touches on foreign affairs, social issues and economic policy with little to no link between these topics can easily be considered as multiple

standalone debates. The specific way in which a televised debate is held, where the topic shift is explicitly conducted by the moderator, whose turn does not impact the debate in any other way, gives the viewer a clear separation of topics that can be used as the segmentation between sub-debates.

Each of the debates was transcribed by one annotator, and corrected by another; a subset of the debates for the second tour of the presidential elections had available transcriptions online[2]. After collecting all of the debates, we asked NLP and political experts to annotate them for multiple tasks. All annotators of the dataset, whether for summary creation or human evaluation, are native french speakers specialized either in computer science or in political science. The following annotations have been conducted on all debates:

1. Abstractive Summarization

2. Extractive Summarization

3. Abstractive Community annotation

Each of the debates contains three different abstractive summaries, two extractive summaries, and two sets of annotations for abstractive communities, as explained below. The data is released on Github

## 2.2 Annotations

In this section, we explain the rules we followed to create each type of annotation.

### 2.2.1 Extractive summaries

An extractive summary is used to condense a longer text into a shorter text that includes only the most important information, used to provide readers with an overview of a longer text without requiring them to read the entire document. An effective extractive summary needs to capture the main points of the text while removing unnecessary details to ensure the summary is as condensed as possible, while still giving context to support the main claims.

While extractive summaries lend themselves well to the summarization of traditional documents, the nature of spontaneous speech, especially for long discussions, leads to a strong preference for abstractive summarization. Indeed, spontaneous speech has many disfluencies, and long conversations tend to have sentences that have repetitions or trail off. The density of information varies greatly,

---

[2]https://www.vie-publique.fr/

and humans tend to prefer abstractive summarization for conversation (Murray et al., 2010).

**Instructions.** Considering that debates, unlike meetings and other long, spontaneous discussions, are held between multiple well prepared speakers, some of the classical problems of extractive summarization for dialogues are diluted (there are relatively few disfluencies and hesitations). However, they are still present. The annotators have been instructed to (1) Create short summaries, (2) extract the most important pieces of information as well as their context; specifically, extract any specific propositions given by a candidate, (3) extract entire sentences, preceded by the speaker's name, (4) extract rhetorical clauses and conclusions, (5) make a summary that is, as much as possible without going against (1), readable as a standalone dialogue.

### 2.2.2 Abstractive Summaries

In contrast to extractive summaries, abstractive summaries are more readable and engaging. On the other hand, they are more complicated to design, as well as to generate.

To construct an abstractive summary, the following steps can be taken: (1) Identify the main points of the text, (2) Write novel sentences that concisely summarize them, (3) Check the text for coherence and readability, as well as completeness.

**Instructions.** Debates often follow a systematic structure. After the moderator opens the subject, the first candidate who replies provides a summary of what the party in place did right/wrong, and then explains what he or she wants to propose — the second one does the same, providing a summary and proposing how best to proceed. After this, the different propositions are discussed in terms of budgeting and feasibility.

The question arises as to what a summary should contain. Should it simply explain the propositions enough to summarize both positions, or should descriptions of rebuttals also be included? Many debates contain heavy ad-hominem subsections, with personal accusations that are not political — should an accurate summary also mention them?

In FREDSum, the annotators were asked to collect the candidates summaries, propositions, and the main counter points given, while removing as much as possible any information unnecessary for describing the general idea of the debate.

Annotators started each summary with a list of the participants, as well as the main theme of the

debate. Then, the annotators made three summaries in three different styles to vary the level of abstractivity. The first type of summary seeks to avoid the problem of coreference where possible, by opting for proper names in place of personal pronouns. The second type is an abstractive summary produced based on an extractive summary, while the third type involves the conception of summaries as natural documents.

### 2.2.3 Abstractive Communities

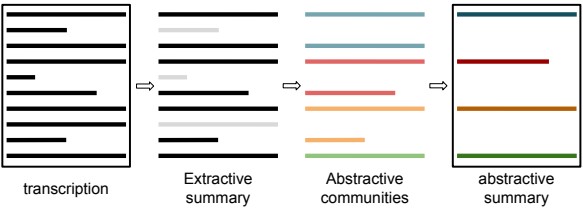

| transcription | Extractive summary | Abstractive communities | abstractive summary |

Figure 1: The creation process of an abstractive summary and abstractive communities. From a source transcription, we create an extractive summary, then group parts of it in communities of extractive sentences supporting a same topic, summarized by one abstractive sentence

In the scope of the dataset, abstractive communities are groupings of extractive sentences in clusters that each support a sentence in the abstractive summary; in this way, each of the sentences of the abstractive summary are supported by one or more of the extractive utterances.

To construct abstractive communities, an NLP system first identifies the key phrases and sentences in the transcript that represent the main ideas and arguments. It then groups these phrases and sentences together based on their semantic similarity, using techniques such as clustering or topic modeling. The resulting groups of phrases represent the abstractive communities.

**Instructions** The instructions for creating abstractive communities are fairly straightforward, we provided the annotators with both the source debate to acquaint themselves with the data, as well as both the extractive and abstractive summary done by each of the annotators. The annotators were then asked to use Alignmeet (Polák et al., 2022) to create the communities.

### 2.3 Pretraining corpora

In order to have efficient systems for summarization or other generational tasks, a large amount of pretraining data is needed. In the scope of French politics, few data are readily available (Koehn, 2005; Ferrara, 2017; Abdine et al., 2022), and only a small part of this data is comprised of dialogues. In the hope of further providing data for pre-training, we propose to release a preprocessed corpus of dialogue in French for pretraining models on French political data, containing all of the senate questions from 1997 as well as the Parliament sessions from 2004 onwards.

To gather the necessary data, we scraped the official websites of the French Senate and Parliament to obtain publicly available datasets. The original data consisted of both HTML pages as well as pdfs that have been processed and formatted to ensure their cleanliness as well as consistency for use in NLP tasks.

The aim of gathering this pre-training data is two-fold. Having the data readily available for pre-training will help with a variety of tasks, such as abstractive summarization of debates, but can also provide a useful resource for further studies on sociological aspects of the evolution of French politics throughout the Fifth Republic, and can therefore be used as a foundation not only for developing NLP models, but also for diverse subjects touching on French politics.

## 3 Statistics

FREDSum is comprised of 18 debates segmented into 138 sub-debates by topic. Each of the sub-topics have two extractive summaries as well as three abstractive summaries. Extractive summaries are naturally longer than their abstractive counterparts; extractive summaries contain an average of 684 tokens while abstractive summaries have an average of 239.

## 4 Experiments

### 4.1 Abstractive summarization

In this section, we aim to evaluate three different representative models on our abstractive summaries. The three models we choose to use are BARThez (Kamal Eddine et al., 2021), Open Assistant (Köpf et al., 2023), and ChatGPT (Ouyang et al., 2022). While ChatGPT and Open Assistant represent models from the newer paradigm of NLP — large language models based on prompting, with Open Assistant relying on LLAMA 30B — we also decide to showcase the results of previous state of the art french models for generation

| Debate |
| --- |
| P1 : Alors, l'autre question euh qui est apparue, et qui était très nette pendant cette campagne, c'est le pouvoir d'achat. Vous en avez fait un des emblèmes de de votre campagne, euh et je viens euh vers vous, peut-être euh Jordan Bardella. [...] |
| JB : Nous le compensons. Le chiffrage, il est public, les français peuvent le voir sur le site de campagne de Marine Le Pen. Un budget de soixante-huit milliards qui est présenté à l'équilibre.[...] |
| OV : Tout d'abord, pardon, je pense qu'il y a un souci. [...] La totalité des prestations sociales versées aux étrangers chaque année c'est neuf milliards d'euros. [...] |
| **Summary** |
| Dans ce débat, Olivier Véran et Jordan Bardella parlent du pouvoir d'achat. Jordan Bardella commence en disant que le budget de son parti est à l'équilibre. [...] Il explique que des économies de seize milliards sont faites en supprimant les aides sociales aux étrangers. |

Table 2: An example of a debate annotation, purple text denotes extractive summary.

with BARThez.

The experiments involving ChatGPT and Open Assistant were conducted using the web interface and minimal prompt engineering. The aim was to utilize the models as regular users would. The prompts were identical for both models, consisting of the debate input and the request to "Summarize this debate between [debate members]." Notably, Open Assistant required the question for summarization to be placed at the end of the prompt, whereas ChatGPT did not have this requirement.

**BARThez** (Kamal Eddine et al., 2021). In the case of BARThez, we want to show the results of traditional encoder-decoder systems to dialogue summarization. BARThez is based on BART (Lewis et al., 2020), and fine tuned for summarization on OrangeSUM. Being based on BART, it can only handle short inputs. For these experiments, we truncated the transcripts into the minimal amount of segments of 512 tokens, and concatenated the output summary for each of the inputs to create the final summary.

**Open Assistant** (Köpf et al., 2023). Open assisstant is an LLM fine-tuned on reddit, stack exchange and books from Project Gutenberg, as well as from many volunteers. It is based on LLAMA 30B. In this benchmark, we used the OA_SFT_Llama_30B_7 model, with the preset of k50-Precise, as well as the default parameters. Considering the low amount of tokens it can take into account (1024), we have followed the same rule as with BARThez for the experimentation. Open Assistant would not give summaries depending on the position of the prompt — thus, without much prompt engineering, we provided it the debate segment with the prompt "Summarize this debate between X,Y".

**ChatGPT** (Ouyang et al., 2022). ChatGPT is an LLM made by OpenAI. In this benchmark, we used the online free web interface based on GPT 3.5, with the May 24th version, to generate the summaries. ChatGPT has a much higher token limit and only few debates had to be separated into sub segments. When they needed to be, the same rule was followed as with BARThez. ChatGPT was simply given a prompt asking "Summarize this debate between X,Y".

We evaluate each of the models with the standard ROUGE metric (Lin, 2004), reporting the $F_1$ scores for ROUGE-1, ROUGE-2 and ROUGE-L. Considering the known shortcomings of ROUGE (Lin, 2004), and its focus on lexical matching, we also propose to evaluate the summaries with BERTScore (Zhang et al., 2020). BERTScore takes into account semantic similarity between the generated and reference summary, complementing ROUGE and its lexical matching.

**ROUGE** (Lin, 2004). The standard metric for both dialogue and general text summarization, scores system produced summaries based purely on surface lexicographic matches with a gold summary. We used a publicly available implementation[3]. Rouge-L is sentence-level.

**BERTScore** (Zhang et al., 2020). BERTScore is a metric used to evaluate the quality of text generation by comparing them to reference texts. BERTScore leverages BERT's contextual embeddings to measure the similarity between the generated text and the reference text at the token level. It takes into account both the precision and recall of the token matches, providing a comprehensive evaluation of the text's quality. We used the original implementation[4], we use bert-base-multilingual-

---

[3] https://github.com/google-research/google-research/tree/master/rouge

[4] https://github.com/Tiiiger/bert_score

|  | R-1 | R-2 | R-L | BERTScore |
|---|---|---|---|---|
| BARThez | 39.38 | 11.77 | 18.31 | 69.74 |
| Open Assistant | 39.20 | 11.31 | 19.00 | 71.85 |
| ChatGPT | **48.93** | **19.61** | **25.60** | **75.30** |

Table 3: ROUGE and BERTScore F1 abstractive summarization scores.

cased, without rescale_with_baseline.

Both for ROUGE and BERTScore, when multiple extractive or abstractive reference summaries exist, we score the candidate with each available reference and returning the highest score.

These results suggest that ChatGPT outperforms both BARThez and Open Assistant in terms of abstractive summarization, demonstrating stronger performance across every evaluation metric.

## 4.2 Extractive summarization

We cast the experiments with our annotated extractive summaries in terms of the *budgeted extractive summarization* problem, i.e., generating a summary with a desired number of words. The budget is defined as the average length of human extractive summaries, which in our case is 684 words. Following common practice (Tixier et al., 2017), we also compare with human abstractive summaries, where the budget is 239 words. Results are shown in Table 4 and 5. As a result of the nature of budgeted summarization, utterances may be truncated to meet the hard overall size constraint, which makes evaluating with Precision, Recall and F1 score by matching integral utterances inappropriate. For this reason, we use the same ROUGE and BERTScore metrics as the previous section.

We experimented with the baselines below ranging from heuristic, text graph, and embedding based unsupervised approaches:

**Longest Greedy** (Riedhammer et al., 2008). A recommended baseline algorithm for facilitating cross-study comparison. The longest remaining utterance is removed at each step from the transcript until the summary size limit is met. We implement this simple baseline.

**Textrank** (Mihalcea and Tarau, 2004). In this approach, each utterance in the transcript is treated as a node within an undirected complete graph. The edges between nodes are assigned weights based on the lexical similarity between the utterances. To generate the summary, we select the top nodes according to their weighted PageRank scores. We

|  | R-1 | R-2 | R-L | BERTScore |
|---|---|---|---|---|
| LongestGre | **68.95** | **54.67** | 33.21 | 78.80 |
| Textrank | 63.31 | 43.89 | 44.29 | 80.69 |
| CoreRankSub | 61.22 | 41.24 | 26.96 | 75.54 |
| BERTExtSum | 64.70 | 45.01 | **44.96** | **80.94** |

Table 4: Extractive summarization results with a budged of 684 words (Average length of extractive summaries)

|  | R-1 | R-2 | R-L | BERTScore |
|---|---|---|---|---|
| LongestGre | 39.74 | 10.54 | 16.45 | 66.60 |
| Textrank | 38.20 | 10.07 | 16.92 | 66.86 |
| CoreRankSub | 36.97 | 10.66 | 15.91 | **67.82** |
| BERTExtSum | **40.38** | **11.16** | **17.71** | 67.67 |

Table 5: Extractive summarization results with a budged of 239 words (Average length of abstractive summaries)

used a publicly available implementation[5].

**CoreRank Submodular** (Tixier et al., 2017). A fully unsupervised, extractive text summarization system that leverages a budgeted submodular maximization framework. It features a term scoring algorithm—CoreRank—based on $k$-core decomposition applied on graph-of-words representation of the text. The original implementation is in R code; we re-implemented in Python. The hyperparameters $\lambda$, $r$, $W$ are set following the optimal values found for the AMI meeting corpus (Mccowan et al., 2005) in the original work.

**BERTExtSum** (Miller, 2019). This approach first embeds input sentences with BERT, and then clusters the embeddings with K-Means, so the closest sentences to the $K$ centroids are selected as extractive summary sentences. To meet our summarization setting budgeted in terms of words, we start from setting $K = 1$ and increase its number until the output summary meets our budget constraint (with cutting - meaning we cut the summary to meet the expected number of words). We used a publicly available implementation[6]. We use a fine-tuned multilingual Sentence-BERT (Reimers and Gurevych, 2019) model[7] as our underlying model, provided in the SentenceTransformers library[8], due to its better performance on semantic textual similarity tasks.

---

[5]https://github.com/summanlp/textrank
[6]https://github.com/dmmiller612/bert-extractive-summarizer
[7]paraphrase-multilingual-MiniLM-L12-v2
[8]https://www.sbert.net/

These results show that BERTExtSum generally outperforms other extractive baselines, for both extractive and abstractive summary budgeting. We can also see that BERTExtSum performs very well when compared to the human annotated extractive summaries on BERTScore and R-L, while Longest-Greedy still outperforms other baselines on R-1 and R-2.

## 4.3 Abstractive community detection

In this section, we report the performance of baseline systems on the task of abstractive community detection, in which utterances in the extractive summary are clustered into groups (abstractive communities), according to whether they can be jointly summarized by a common sentence in the abstractive summary. In our annotated dataset, there are in total 123 available pairs of (Extractive 1, Abstractive 1) summaries provided with such linking annotations, and 95 for (Extractive 2, Abstractive 2) summaries. We report their results separately in Table 6 and Table 7.

Following previous work (Murray et al., 2012; Shang et al., 2018, 2020), we experiment with a simple unsupervised baseline by applying $k$-means algorithm on utterance embeddings, obtained by a variety of text embedding methods, such as TF-IDF, Word2Vec[9] (Fauconnier, 2015), multiligual Sentence-BERT (same as the one use in the previous section), and French BERT (FlauBERT[10] (Le et al., 2020)). To account for randomness, we average results over 10 runs.

We compare our $k$-means community assignments to the human ground truth using the Omega Index (Collins and Dent, 1988), a standard metric for this task. The Omega Index evaluates the degree of agreement between two clustering solutions based on pairs of objects being clustered. We report results with a fixed number of communities $k$=11/12, which corresponds to the average number of ground truth communities per extractive-abstractive linking, and also a variable $k$ equal to the number of ground truth communities (denoted by $k$=v). We adopt a publicly available metric implementation[11].

The overall low omega index scores in the results indicate the difficulty of this task, which is consistent with findings in previous work. Results also

[9] frWac_non_lem_no_postag_no_phrase_500_skip_cut100.bin
[10] flaubert_large_cased
[11] https://github.com/isaranto/omega_index

| OmegaI | TF-IDF | Word2Vec | mBERT | frBERT |
|---|---|---|---|---|
| $k$ = v | 11.41 | 10.64 | 12.27 | **20.20** |
| $k$ = 11 | 9.6 | 9.3 | 10.68 | **13.26** |

Table 6: Omega index × 100 of $k$-means algorithm for detecting communities on the extractive-abstractive linking data between extractive 1 and abstractive 1 summaries.

| OmegaI | TF-IDF | Word2Vec | mBERT | frBERT |
|---|---|---|---|---|
| $k$ = v | 13.25 | 12.07 | 10.71 | **17.14** |
| $k$ = 12 | 11.92 | 9.6 | **12.58** | 11.77 |

Table 7: Omega index × 100 of $k$-means algorithm for detecting communities on the extractive-abstractive linking data between extractive 2 and abstractive 2 summaries.

demonstrate the superior performance of BERT based text embeddings across different settings.

## 5 Human evaluation

While ROUGE is still considered as the go-to metric for evaluating summaries, it has inherent limitations that makes it unreliable when used as the sole measure for the assessment of a summary's quality. Indeed, ROUGE focuses on measuring lexical overlaps between reference summaries and generated summaries, lacking the ability to evaluate semantic similarity, or hallucinations. In this paper, we propose to conduct human analysis to be able to evaluate how well both ROUGE and BERTScore scale with human judgment in the case of dialogue summarization, as well as to see how well summaries produced by newer LLMs score with respect to human evaluation. We asked the annotators to rate 50 debates from 1-5 in informativity, which requires the human to assess how well the summary captures the essence and context of the original debate; readability, which will further explore the preference of abstractive summaries people tend to have for dialogues; and faithfulness, which aims at evaluating how much of the generated summary is comprised of made up facts and hallucinations.

As language models advance and become more sophisticated, their summary outputs also may not scale proportionally with ROUGE scores. New techniques, such as controlled generation and reinforcement learning methods, allow these models to produce diverse and high-quality summaries that cannot be adequately captured by ROUGE alone.

|              | Readability | Informativity | Faithfulness |
|--------------|-------------|---------------|--------------|
| Abstractive 1 | 2.97 | 3.87 | 4.96 (2) |
| Abstractive 2 | **4.32** | **4.36** | 4.98 (1) |
| Abstractive 3 | 3.79 | 3.66 | 4.98 (1) |
| Extractive 1 | 2.60 | 4.22 | **5.00** (0) |
| Extractive 2 | 2.63 | **4.37** | **5.00** (0) |
| BARThez | 1.69 | 1.30 | 1.80 (48) |
| ChatGPT | 3.76 | 3.87 | 4.50 (19) |
| OpenAssistant | 3.50 | 2.54 | 2.61 (40) |

Table 8: Human evaluation of Readability, Informativity and Faithfullness out of 5. A high readability score means an easily readable summary, high informativity indicates an exhaustive summary, and high faithfullness means the summary is consistent with the transcript; the number in parentheses is the amount of summaries that contain hallucinations.

Human evaluation provides a complementary assessment to identify the strengths and weaknesses of these advanced models and compare their performance in terms of faithfulness, readability, and informativity.

Multiple points are brought out by the human evaluations. While human annotated summaries consistently rank the highest in all metrics, ChatGPT produces remarkably competitive summaries, particularly in terms of informativity and readability, while maintaining a high level of faithfulness. It is worth noting that, in the case of ChatGPT, although some of the annotated summaries contained hallucinations, these hallucinations differed significantly from those generated by OpenAssistant or BARThez and were more similar to what human annotators would produce. These hallucinations typically resulted from either ambiguous speech in the original transcript or an attempt to capture the original tone of a speaker, such as mentioning irony when it does not exist.

Furthermore, although BARThez appeared relatively competitive in terms of ROUGE and BERTScore, its summaries were considered significantly worse when compared to human judgment.

Lastly, while extractive summaries may rank lower in readability due to the nature of dialogue speech, on average, they outperform abstractive summaries in both informativity and faithfulness. Additionally, it is challenging to include hallucinated facts in extractive summaries, as doing so would require omitting negation clauses entirely. For tasks where faithfulness is crucial, extractive summaries continue to rank higher than abstractive ones. The higher informativity of extractive summaries is likely a result of their more extensive content.

# 6 Meta-evaluation of automatic metrics

As we have mentioned previously, the automatic evaluation metrics used in summarization have limitations that prevent them from perfectly assessing the quality of a given model. In this subsection, we aim to determine the correlation between widely used metrics such as ROUGE-1, ROUGE-2, ROUGE-L, and BERTScore, and human judgment. For the sake of comprehensiveness, we employ the grading scheme discussed in section 5, encompassing Readability, Informativity, and Faithfulness.

|           | Readability | Informativity | Faithfulness |
|-----------|-------------|---------------|--------------|
| Rouge-1 | 0.22 | 0.54 | 0.52 |
| Rouge-2 | 0.24 | 0.64 | 0.59 |
| Rouge-L | 0.37 | **0.69** | **0.64** |
| BERTScore | **0.53** | **0.69** | **0.64** |

Table 9: Pearson correlation between human judgement and automatic metrics in Readability, Informativity and Faithfulness

The results show that, as expected, ROUGE-1 performs worse of all available metrics across the board. On the other hand, both BERTScore and ROUGE-L demonstrate noteworthy Pearson correlations, particularly in terms of Informativity and Faithfulness. Notably, BERTScore consistently exhibits stronger correlations across all three aspects (Readability, Informativity, and Faithfulness).

# 7 Usage of National Assembly and Senate data

All the data provided in this paper follows the terms and conditions of their respective sources; National Assembly, Senate, following article L. 122-5 of the french intellectual property code, stipulating that there are no copyright for any of the articles, allowing people to copy them freely. The caveat being that the usage has to be either personal, associative or for professional uses, forbidding reproduction for commercial and advertisement purposes

# 8 Conclusion

In this paper, we present the first version of FRED-Sum, a French dataset for abstractive debate summarization. The dataset consists of manually transcribed political debates, covering a range of topics

and perspectives and annotated with topic segmentation, extractive and abstractive summaries and abstractive communities. We conduct extensive analyses on all the annotations and provide baseline results on the dataset by appealing to both automatic and human evaluation.

## Limitations

In this section, we discuss several limitations on the dataset and baseline used in this paper.

- Limited amount of data: Although this work represents the first resource of French summarized debate data, or dialogue data in general, the total duration of 50 hours is relatively low; this inevitably limits the impact o the dataset for training efficient systems for generalized dialogue summarization. A longer dataset would enhance the robustness and generalizability of the findings

- Minimal work on prompt engineering: The nature of the experiments, as well as the need for replicability, made it so that both of the prompt based models were not used to their full capabilities, as little prompt engineering has been conducted. This may have hindered the model's ability to adapt and to generate contextually appropriate responses. Further exploration of prompt engineering could lead to improved results comparatively.

- Lack of linguistic annotations: Many different kinds of systems have been used for dialogue summarization, using a wide variety of annotations to help with the initial interpretation of a transcript, to offset the lengthy input issue. While there is interest in providing abstractive and extractive summaries, as well as abstractive communities, some models require dialogue act annotations as well as discourse graphs to function. Providing further annotations would allow more traditional systems to be tested on this dataset.

- Limited fine-tuning: The model used in this study, Barthez, has not undergone extensive fine-tuning specifically for the French debate domain, as well as for conversations in general. While Barthez is a powerful language model, the lack of fine-tuning tailored to debate-related tasks may impact its performance and limit its ability to capture nuanced arguments and the intricacies of domain-specific language.

## Ethical Consideration

Considering the nature of any political paper, ethics is a concern. We certify that any data provided in this paper follows the terms and conditions of their respective sources; National Assembly, Senate. Moreover all human annotations have been gathered in a fair fashion, each annotators being fully compensated for the annotations as part of their full time job at our organization, as well as being authors or acknowledged members of the paper. Finally, we have ensured that the french political landscape is fairly represented in both the pretraining dataset; all presidential debates have been taken into account and every political proximity that has gotten at least 5% of the votes at a presidential election is present in at least one debate as well as in the choices of debates summarized. In addition, it is worth noting that the annotators openly represent diverse political affiliations so as to average any bias there might be in summaries, however, bias is inevitable in any such task, we hope that having multiple source for summaries helps lowering the average bias.

## Acknowledgements

This work was supported by both the SUMM-RE (ANR-20-CE23-0017) and the Horizon Europe CORTEX² (CL4-2021-HUMAN-01-25) projects. We thank Rayan Autones and Jules Peyrat for contributing in the meta-evaluation research of automatic metrics, during their internship.

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

# A  Additional experiments

In the following appendix, we provide supplementary information that complements the main text of this paper. This includes detailed methodologies, experimental setups, and additional results not presented in the main body. The intention of this appendix is to offer readers a deeper understanding of the experiments conducted, as well as to provide transparency regarding our research processes.

**Inter-annotator agreement** : Defining inter-annotator agreement in the context of extractive summarization can be challenging due to the repetition of identical utterances in transcripts, requiring both annotators to select the same portions. Nevertheless, we can approximate it by computing the ROUGE and BERTScore scores between extractive summaries.

|  | R1 | R2 | RL | BERTScore |
|---|---|---|---|---|
| Ext 1 vs Ext 2 | 0.72 | 0.59 | 0.69 | 0.95 |

Table 10: ROUGE and BERTScore comparison of both extractive summaries

**Human Evaluation criterias**  Human evaluations play a pivotal role in assessing the effectiveness of the methods as well as the quality of the data. While automated metrics give a quantitative measure, human judgments provide qualitative insights that are often more aligned with practical use-cases, in this case, we gave the following guidelines for the annotators when rating for faithfullness, Informativity and Readability.

- *Readability* :

1. The given summary is incomprehensible
2. An overall theme can be gathered from reading the summary, but the sentences are heavy and/or repetitive
3. The summary is readable with little repetition, an the point come across easily
4. The summary reads fluently
5. The summary reads fluently, and is elegant

- *Informativity* :

1. The summary captures no information from the original debates
2. The summary captures the general theme of the original debate (The speakers and subject)
3. The summary captures the general theme of the original debate, as well as the main points made by the different debaters
4. The summary captures the general theme of the original debate, the main points made by the different speakers, as well as the responses from the opposite parties
5. The summary perfectly captures the nuances of the original debate

- *Faithfulness* :

1. The summary hallucinates the entirety of the debate.
2. Major hallucinations, such as mistaking the participants of the debates or the main theme of the debate.
3. The summary is faithful enough to be read, but has some minor fabrications, a point made by a speaker is however modified enough to introduce confusion.
4. The summary is largely faithful to the original debate, and only includes hallucinations that could be made by a human because of ambivalent wordings in the original transcript
5. The summary is entirely faithful

Additionally, we propose to give a categorization of hallucinations, from the worse cases that causes complete misunderstanding of the original document to more ambiguous mistakes

- *Complete invention* : The summary starts and develops a subject completely absent from the original transcript; as an example : "Tonight in * emission name *, Jean Luc Mélenchon is interviewed by Laurent ruquier" - In this case, the transcript mentions niether the show nor either of the people cited in the summary. Barthez is prone to those hallucinations

- *Confusion of scale or speaker* : Either attributing a decision to the wrong speaker, or exagerating its scale - "Jordan Bardella proposes to build 35000 prison spots" can be wrong in two ways here: the amount of spots, or the fact that it was another debater that proposed to build the 35000 places - Open Assistant tends to make this kind of mistake.

- *Unavailable information - **correct or not*** : LLMs use information about different characters that are not available in the transcript - This usually appears when a speaker is introduced, such as in the example: "Jordan Bardella, président du rassemblement national." These instances of adding information not directly present in the text but in the pretraining data can exhibit varying degrees of inaccuracies. While it is arguably already a hallucination when the title is correct, these instances can take two other forms: 1) Inaccurate titles, 2) Once correct titles that are no longer current. It's worth noting that some speakers might switch parties or titles, participating in a debate while holding a position of a deputy, yet the summarization process might attribute a different title from a different time period. This tendency towards introducing such hallucinations is more pronounced in ChatGPT, likely stemming from the large pretraining it underwent, although all three models share this characteristic.

- *Triple coreference and confusing speech* : Some parts of debates, with many interruption and references to facts can be complicated to follow : "About that, with the age of the retirement, we will raise it", will often be summarized about raising the age of retirement, while in context - this is about raising the amount of years of contribution - this is often caused, while being correct, by the overlapping of multiple coreferences. - Human hallucinations also fall in this category

- *Irony* : One orator can ironically agree with a

proposition of another while in fact disagreeing with it - ChatGPT can fail to pick up on this kind of nuance.

We note that while no hallucinations, or rather, misunderstandings, are made in the extractive summaries here, it is because we have instructed the annotators to explicitly get rhetorical clauses and context in the extractive summaries (somewhat justifying the lengthy nature of our extractive summaries) - Systems that create extractive summaries, as well as human annotated extractive summaries that do not gather context can contain hallucinations (Zhang et al., 2023).

**Additional experiments for abstractive summarization** : In the experiments we conduct, we compare the models with a specific truncating strategy to optimize their respective outputs and so that all the models get to see all of the debate. To compare as fairly as possible the models, we propose to give additional result in which all models only get to see the largest chunk of the original debate as they can. Moreover, we propose to share the performance of models against each of the abstractive summaries, to see the impact the style of each summary has on the results.

|  | R-1 | R-2 | R-L | BERTScore |
|---|---|---|---|---|
| BARThez | 14.54 | 04.87 | 09.53 | 64.82 |
| Open Assistant | 29.11 | 07.31 | 15.21 | 69.68 |
| ChatGPT | 44.63 | 17.22 | 24.34 | 74.54 |

Table 11: ROUGE and BERTScore F1 abstractive summarization scores against Abstractive 1 - No segmentation.

|  | R-1 | R-2 | R-L | BERTScore |
|---|---|---|---|---|
| BARThez | 15.24 | 04.91 | 10.22 | 64.23 |
| Open Assistant | 30.06 | 07.45 | 16.63 | 69.91 |
| ChatGPT | 42.11 | 17.10 | 23.85 | 73.88 |

Table 12: ROUGE and BERTScore F1 abstractive summarization scores against Abstractive 2 - No segmentation.

|  | R-1 | R-2 | R-L | BERTScore |
|---|---|---|---|---|
| BARThez | 13.28 | 03.76 | 08.82 | 63.06 |
| Open Assistant | 29.07 | 07.45 | 14.11 | 68.66 |
| ChatGPT | 43.22 | 16.51 | 24.19 | 73.67 |

Table 13: ROUGE and BERTScore F1 abstractive summarization scores against Abstractive 3 - No segmentation.