# OpenReview forum: "FREDSum: A Dialogue Summarization Corpus for French Political Debates"
_EMNLP/2023/Conference — EMNLP 2023 Findings_

### Official Review · Reviewer_jbix · 2023-08-03

**Typos Grammar Style And Presentation Improvements:** A typo in Table 8, “Faitfullness” sho…
**Soundness:** 2

**Excitement:**

3: Ambivalent: It has merits (e.g., it reports state-of-the-art results, the idea is nice), but there are key weaknesses (e.g., it describes incremental work), and it can significantly benefit from another round of revision. However, I won't object to accepting it if my co-reviewers champion it.

**Paper Topic And Main Contributions:**

The paper presents a novel dialogue summarization dataset in French on political debates, in the context of a lack of non-English data resources of this kind. The paper collects data from open source and annotate transcripts with multi-granularity human-written summaries. The authors also conduct preliminary experiments benchmarking some models on abstractive and extractive summarization tasks against the dataset.

**Questions For The Authors:**

1.	The study mention employing “a NLP system” identify key phrases and cluster them into sentence groups by topic (line 263). Could author explain the system and any related quality control measures? Since the accuracy of such automatic grouping may significantly affect the quality of following summary annotation procedures.

2.	What do “preS” and “preA”  subscripts in Table 1 stand for? Also, the # token per transcript is unusually small (2595.5) for a meeting transcript, do the transcripts here are the after-extracted ones rather than full transcripts?

**Reasons To Accept:**

The study presents the research community with a novel data in dialogue summarization. Notably, the dataset provides human-written abstractive and extractive summaries with different levels of abstractivity and semantic aspects.

**Reasons To Reject:**

While the scope and characteristic of the dataset proposed is specific, the size of the dataset (18 debates consisting of 138 sub-debates), as the author point out in the Limitation section, is relatively small.

In the experiment, the comparability of the performance of the three models on abstractive summarization is questionable since the models were provided with source transcripts with different truncating strategies (transcripts were truncated by 512 for BARThez, 1024 for LLAMA 30, and mostly untruncated for GPT3.5).

Moreover, the dataset contains two extractive and three different abstractive summaries for each sub-debate, and the study “score the candidate with each available reference and returning the highest score”, which is likely comparability-wise unsound.

**Reproducibility:**

3: Could reproduce the results with some difficulty. The settings of parameters are underspecified or subjectively determined; the training/evaluation data are not widely available.

**Reviewer Confidence:**

4: Quite sure. I tried to check the important points carefully. It's unlikely, though conceivable, that I missed something that should affect my ratings.

---

> ### Author Rebuttal · Authors · 2023-08-28
>
> We would like to thank you for your detailed and insightful review. We respond point-by-point to your comments below
>
> **Reasons To Accept**:
> The study presents the research community with a novel data in dialogue summarization. Notably, the dataset provides human-written abstractive and extractive summaries with different levels of abstractivity and semantic aspects.
>
>
> *We thank the reviewer for acknowledging that our dataset is a novel contribution as the first French dialogue summarization dataset, and provides multiple types of human annotations. We would like to resolve several misunderstandings.*
>
> ---------------------------------------------------
>
>
> **Reasons To Reject:**
> While the scope and characteristic of the dataset proposed is specific, the size of the dataset (18 debates consisting of 138 sub-debates), as the author point out in the Limitation section, is relatively small.
>
> *This can be explained by the large time requirement for producing such data. While the size of our dataset is limited (138 sub-debates), it is on the same level as previous similar corpus, such as AMI (137 meetings) and ICSI (75 meetings), of which the small quantity has not prevented them from being widely used in related areas of research.
> Additionally, we need to recall that for each dialogue transcription, we provide multiple annotations of 3 abstractive summaries, 2 extractive summaries, and 2 abstractive communities, while to the best of our knowledge, no previous dataset proposes more than one annotation for each dialogue.
> In the end, apart from the annotated set, we also provide an unannotated set of debate transcriptions (FREDSum_preS and FREDSum_preA), consisting of more than 7000 debates, each longer than two hours. We believe these two sets are useful for the continual pre-training and Instruction-tuning of a LLM, to construct a system without requiring a significant amount of human annotations.*
>
> ---------------------------------------------------
>
> **Reasons To Reject:** In the experiment, the comparability of the performance of the three models on abstractive summarization is questionable since the models were provided with source transcripts with different truncating strategies (transcripts were truncated by 512 for BARThez, 1024 for LLAMA 30, and mostly untruncated for GPT3.5).
>
> *The logic behind these experiments was to be able to reflect the best performance of each model and using them as would someone who had to summarize the debates with each of those systems. We have however run additional experiments to reflect the results obtained by following an alternative truncating strategy in which all models get as an input the longest string they can compute that does not cut an utterance. Expectedly, the results for Barthez and Open assistant both suffer from this strategy, but this does allow a fairer comparison between the models*
>
>
>
> *All models - No segmentation vs Abstractive 1 :*
> |                | R1   | R2   | RL   | BERTScore |
> |----------------|------|------|------|-----------|
> | ChatGPT        | 0.44 | 0.17 | 0.24 | 0.74      |
> | Open Assistant | 0.29 | 0.07 | 0.15 | 0.69      |
> | Barthez        | 0.14 | 0.04 | 0.09 | 0.64      |
>
>
> *All models - No segmentation vs Abstractive 2 :*
> |                | R1   | R2   | RL   | BERTScore |
> |----------------|------|------|------|-----------|
> | ChatGPT        | 0.42 | 0.17 | 0.23 | 0.73      |
> | Open Assistant | 0.30 | 0.07 | 0.16 | 0.69      |
> | Barthez        | 0.15 | 0.04 | 0.10 | 0.64      |
>
>
>
> *All models - No segmentation vs Abstractive 3 :*
> |                | R1   | R2   | RL   | BERTScore |
> |----------------|------|------|------|-----------|
> | ChatGPT        | 0.43 | 0.16 | 0.24 | 0.73      |
> | Open Assistant | 0.29 | 0.07 | 0.14 | 0.68      |
> | Barthez        | 0.13 | 0.03 | 0.08 | 0.63      |
>
>
>
>
>
>
>
> ---------------------------------------------------
>
> **Reason to reject :** Moreover, the dataset contains two extractive and three different abstractive summaries for each sub-debate, and the study “score the candidate with each available reference and returning the highest score”, which is likely comparability-wise unsound.
>
> *When multiple references exist, it is the standard practice to first score the candidate summary with respect to each reference summary, and then keep the highest score as the final one. Please refer to the original ROUGE paper [1] (Section 2.1 Multiple References) and BERTScore paper [2] (page 6), as well as the metric implementations listed in the footnotes of our paper.*
>
> *[1] Lin, C. Y. (2004, July). Rouge: A package for automatic evaluation of summaries. In Text summarization branches out (pp. 74-81).*
>
> *[2] Zhang, T., Kishore, V., Wu, F., Weinberger, K. Q., & Artzi, Y. (2019). Bertscore: Evaluating text generation with bert. arXiv preprint arXiv:1904.09675.*
>
> *Additionally, we propose to report below results obtained by each system against each abstractive and extractive summaries. Results show that the scaling of the scores stay the same with each evaluation strategies on abstractive summaries*
>
>
> *All models vs Abstractive 1 :*
> |                | R1   | R2   | RL   | BERTScore |
> |----------------|------|------|------|-----------|
> | ChatGPT        | 0.45 | 0.18 | 0.24 | 0.74      |
> | Open Assistant | 0.36 | 0.10 | 0.18 | 0.70      |
> | Barthez        | 0.36 | 0.10 | 0.17 | 0.68      |
>
>
>
> *All models vs Abstractive 2 :*
> |                | R1   | R2   | RL   | BERTScore |
> |----------------|------|------|------|-----------|
> | ChatGPT        | 0.43 | 0.17 | 0.23 | 0.74      |
> | Open Assistant | 0.36 | 0.09 | 0.18 | 0.71      |
> | Barthez        | 0.37 | 0.10 | 0.17 | 0.67      |
>
>
>
> *All models vs Abstractive 3 :*
> |                | R1   | R2   | RL   | BERTScore |
> |----------------|------|------|------|-----------|
> | ChatGPT        | 0.47 | 0.16 | 0.23 | 0.73      |
> | Open Assistant | 0.37 | 0.09 | 0.17 | 0.70      |
> | Barthez        | 0.37 | 0.08 | 0.16 | 0.68      |
>
> *All the results above will be provided in an appendix for the final version of the paper*
>
> ---------------------------------------------------
>
> **Questions for the authors :** The study mention employing “a NLP system” identify key phrases and cluster them into sentence groups by topic (line 263). Could author explain the system and any related quality control measures? Since the accuracy of such automatic grouping may significantly affect the quality of following summary annotation procedures.
>
> *Abstractive communities are obtained via human annotations of alignment between extractive summary and abstractive summary. We give two pairs of abstractive communities for each of the 138 sub topics*
>
> *Additionally we will refine Section 2.2.3 by briefly introducing the two previous works on the sub-task of abstractive community detection. To find utterance clusters, one [1] uses sentence graph + community detection (CONGA) algorithm, another [2] uses sentence embeddings (obtained with siamese network) + Fuzzy-C means algorithm. The answers to the other questions can be found in Section 4.3, we use Omega Index to evaluate the performance.*
>
> *[1] Murray, G., Carenini, G., & Ng, R. (2012, June). Using the omega index for evaluating abstractive community detection. In Proceedings of Workshop on Evaluation Metrics and System Comparison for Automatic Summarization (pp. 10-18).*
>
> *[2] Shang, G., Tixier, A. J. P., Vazirgiannis, M., & Lorré, J. P. (2019). Energy-based self-attentive learning of abstractive communities for spoken language understanding. arXiv preprint arXiv:1904.09491.*
>
>
> ---------------------------------------------------
>
> **Questions for the authors :** What do “preS” and “preA” subscripts in Table 1 stand for? Also, the # token per transcript is unusually small (2595.5) for a meeting transcript, do the transcripts here are the after-extracted ones rather than full transcripts?
>
> *PreS represents the statistics of the senate subsate of the pretraining data, while PreA represents the assembly*
>
> *The transcripts here refers to full transcripts, while the #tokens here actually represents the number of words; the number still being fairly low as some sub debates can be very short - Clarification for both of those points will be added to the paper*
>
> ---------------------------------------------------
>
> *Thank you again for the review, we hope to have cleared up the points presented in your reasons to reject*

---

### Official Review · Reviewer_Xx4E · 2023-08-05

**Soundness:** 3

**Ethical Concerns:**

Yes

**Excitement:**

3: Ambivalent: It has merits (e.g., it reports state-of-the-art results, the idea is nice), but there are key weaknesses (e.g., it describes incremental work), and it can significantly benefit from another round of revision. However, I won't object to accepting it if my co-reviewers champion it.

**Justification For Ethical Concerns:**

The authors scraped the official websites of the French Senate and Parliament to create a publicly available pre-training dataset in French. What were the content use-policies of these websites? Is creation of such dataset and training of models on such data allowed under this policy? If the curated dataset violates any terms of use, then this could be a major concern. While the authors mention briefly in the ethics section that the any data provided in this paper follows the terms and conditions of their respective sources, it would be important to explicitly state the allowed usage of this dataset (what is allowed and what is not) in the paper itself. For instance, commercial use of any information is not allowed, which should be mentioned in the paper to avoid use of such dataset for training any industrial solutions etc. This is only an example, more thorough understanding the policy use is required and need to be discussed in the paper.

**Paper Topic And Main Contributions:**

This paper provides a resource for dialouge summarization in French. In particular, the authors curate a dataset of human-transcribed political dialogues in French and provide human-annotated summaries (both extractive and abstractive). The authors also provide alignment annotations between extractive and abstractive summary sentences, referred as abstractive communities. The authors further experiment with several existing summarization models and alignment approaches (based on clustering) and demonstrate the performance of current systems on this task.

**Questions For The Authors:**

1. Line 469: Explaining what is meant by 'with cutting' could help readers.
2. Lines 266-278: It is unclear if the abstractive communities are obtained via clustering/topic modeling or human annotations of alignment between extractive and abstractive summary sentences. A clarification can help readers.
3. Choice of 684 and 249 words for length-constrained summaries can be explained. Why these choices are reasonable to make? The authors provide citations for such choice, but describing in a line or two in the paper itself could help readers.
4. Line 571-575 It would be interesting to provide examples of hallucinations and ironies the authors describe in the paper. Error analysis could be in general interesting to researchers interested in using this resource. While I do not advocate for English only, providing an English translation of the French examples could help a wider audience who does not speak French and thus enhance the impact of the paper.
5. Line 592: It is unclear what is the motivation to study the correlation between human annotations and automatic metrics. Such studies have been conducted by the authors of automated metrics, and it is unclear how it is contextualized in this work. More explanation of insights drawn based on this analysis, which is specifically applicable to this paper, could be useful.



**Reasons To Accept:**

1. The authors provide a valuable resource of human-transcribed French political debates with human-annotated summaries. This resources can not only be useful for summarization task but also to employ computational methods to study French politics.
2. The authors also provide several baseline experiments demonstrating the current capabilities of summarization models.

**Reasons To Reject:**

1. Line 292: The authors scraped the official websites of the French Senate and Parliament to create a publicly available pre-training dataset in French. What were the content use-policies of these websites? Is creation of such dataset and training of models on such data allowed under this policy? If the curated dataset violates any terms of use, then this could be a major concern. While the authors mention briefly in the ethics section that the "any data provided in this paper follows the terms and conditions of their respective sources," it would be important to explicitly state the allowed usage of this dataset (what is allowed and what is not) in the paper itself. For instance, commercial use of any information is not allowed, which should be mentioned in the paper to avoid use of such dataset for training any industrial solutions etc. This is only an example, more thorough understanding of the terms of use is required and need to be discussed in the paper.

2. The authors provide baseline experiments for extractive, abstractive, and abstractive community detection. However, no discussion is provided on what are the research gaps addressed by this dataset. Based on the provided experiments, some of the models perform fairly well, which is interesting to note in itself. A discussion of where do current models already do well and where do they lack could be useful for future researchers interested in using this dataset as a resource.

**Reproducibility:**

3: Could reproduce the results with some difficulty. The settings of parameters are underspecified or subjectively determined; the training/evaluation data are not widely available.

**Reviewer Confidence:**

4: Quite sure. I tried to check the important points carefully. It's unlikely, though conceivable, that I missed something that should affect my ratings.

**Typos Grammar Style And Presentation Improvements:**

Line 428 Rrcall-->Recall
Table 5 is for 239 words?

---

> ### Author Rebuttal · Authors · 2023-08-28
>
> We would like to thank you for your detailed and insightful review. We respond point-by-point to your comments below
>
> ------------------------------------------
>
> **Reasons To Accept:**
> The authors provide a valuable resource of human-transcribed French political debates with human-annotated summaries. This resources can not only be useful for summarization task but also to employ computational methods to study French politics.
> The authors also provide several baseline experiments demonstrating the current capabilities of summarization models.
>
> *We thank the reviewer for confirming that our dataset is a valuable resource for dialogue summarization with multiple types of annotation, and it additionally opens up opportunities for novel research in computational methods to study French politics.*
>
> ------------------------------------------
>
> **Reason to reject:** Line 292: The authors scraped the official websites of the French Senate and Parliament to create a publicly available pre-training dataset in French. What were the content use-policies of these websites? Is creation of such dataset and training of models on such data allowed under this policy? If the curated dataset violates any terms of use, then this could be a major concern. While the authors mention briefly in the ethics section that the "any data provided in this paper follows the terms and conditions of their respective sources," it would be important to explicitly state the allowed usage of this dataset (what is allowed and what is not) in the paper itself. For instance, commercial use of any information is not allowed, which should be mentioned in the paper to avoid use of such dataset for training any industrial solutions etc. This is only an example, more thorough understanding of the terms of use is required and need to be discussed in the paper.
>
> *This is an excellent point; an additional section will be added to the paper in order to explain the licensing more thoroughly; Both the National Assembly and the Senate transcripts follow article L.122-5 of the french intellectual property code, stipulating that there are no copyright for any of the articles, allowing people to copy them freely. The caveat being that the usage has to be either personal, associative or for professional uses, forbidding reproduction for commercial and advertisement purposes*
>
> *As such, the creation of a dataset as well as the training of models in the scope of research is allowed*
>
> ------------------------------------------
>
> **Reason to reject:** The authors provide baseline experiments for extractive, abstractive, and abstractive community detection. However, no discussion is provided on what are the research gaps addressed by this dataset. Based on the provided experiments, some of the models perform fairly well, which is interesting to note in itself. A discussion of where do current models already do well and where do they lack could be useful for future researchers interested in using this dataset as a resource.
>
> *As evident from the human assessment of summaries, ChatGPT performs well in abstractive summarization, contrary to models such as Open Assistant and Barthez, who display notable shortcomings across all fronts. Although Open Assistant demonstrates relative informativity in managing concise debates, the prevalence of hallucinations and the absence of exhaustive summaries highlight the significant challenges that open-source models still face. Moreover, despite ChatGPT's strong performance, it still has limitations in terms of informativity and faithfulness. This becomes particularly apparent in longer debates and discussions with frequent interruptions or disfluencies. Furthermore, there is a necessity for continued efforts in extractive summarization within the context of dialogues, alongside advancements in abstractive community detection.*
>
>
> ------------------------------------------
>
> **Questions for the authors:** Line 469: Explaining what is meant by 'with cutting' could help readers.
>
> *With our described way of using BERTExtSum, the resulting extractive summary may be longer than the budget constraint, “with cutting” here simply means we cut the summary to meet the expected number of words. Clarification will be added to the paper*
>
> ------------------------------------------
>
> **Questions for the authors:** Lines 266-278: It is unclear if the abstractive communities are obtained via clustering/topic modeling or human annotations of alignment between extractive and abstractive summary sentences. A clarification can help readers.
>
> *Abstractive communities are obtained via human annotations of alignment between one extractive summary and one abstractive summary. Each of the 138 sub debates thus has two different annotations for abstractive communities*
>
> ------------------------------------------
>
> **Questions for the authors:** Choice of 684 and 249 words for length-constrained summaries can be explained. Why these choices are reasonable to make? The authors provide citations for such choice, but describing in a line or two in the paper itself could help readers.
>
> *Using the average length of summaries for budgetting is usual, we will mention this further in the paper*
>
> ------------------------------------------
>
> **Questions for the authors:** Line 571-575 It would be interesting to provide examples of hallucinations and ironies the authors describe in the paper. Error analysis could be in general interesting to researchers interested in using this resource. While I do not advocate for English only, providing an English translation of the French examples could help a wider audience who does not speak French and thus enhance the impact of the paper.
>
> *Different level of hallucinations can be observed of varying degrees of importance :*
>
> - Complete invention : The summary starts and develops a subject completely absent from the original transcript; as an example : “Tonight in * emission name *, Jean Luc Mélenchon is interviewed by Laurent ruquier” - In this case, the transcript mentions niether the show nor either of the people cited in the summary. Barthez is prone to those hallucinations
> - Confusion of scale or speaker - Either attributing a decision to the wrong speaker, or exagerating its scale - “Jordan Bardella proposes to build 35000 prison spots” can be wrong in two ways here: the amount of spots, or the fact that it was another debater that proposed to build the 35000 places - Open Assistant tends to make this kind of mistake.
> - Unavailable information - *correct or not* : LLMs use information about different characters that are not available in the transcript - This usually appears when a speaker is introduced, such as in the example: "Jordan Bardella, président du rassemblement national." These instances of adding information not directly present in the text but in the pretraining data can exhibit varying degrees of inaccuracies. While it is arguably already a hallucination when the title is correct, these instances can take two other forms: 1) Inaccurate titles, 2) Once correct titles that are no longer current. It's worth noting that some speakers might switch parties or titles, participating in a debate while holding a position of a deputy, yet the summarization process might attribute a different title from a different time period. This tendency towards introducing such hallucinations is more pronounced in ChatGPT, likely stemming from the large pretraining it underwent, although all three models share this characteristic.
> - Triple coreference and confusing speech - Some parts of debates, with many interruption and references to facts can be complicated to follow : “About that, with the age of the retirement, we will raise it”, will often be summarized about raising the age of retirement, while in context - this is about raising the amount of years of contribution - this is often caused, while being correct, by the overlapping of multiple coreferences. - Human hallucinations also fall in this category
> - Irony - One orator can ironically agree with a proposition of another while in fact disagreeing with it - ChatGPT can fail to pick up on this kind of nuance.
>
> *Finally, it is important to note that while there are no hallucinations made in the extractive summaries here, it is because we have instructed the annotators to explicitly get rhetorical clauses and context in the extractive summaries (somewhat justifying the lengthy nature of our extractive summaries) - Systems that create extractive summaries, as well as human annotated extractive summaries that do not gather context can contain hallucinations [1]*
>
> *[1] Zhang, Shiyue, David Wan and Mohit Bansal. "Extractive is not faithful: An investigation of broad unfaithfulness problems in extractive summarization." arXiv preprint arXiv:2209.03459 (2022).*
>
>
> ------------------------------------------
>
> **Questions for the authors :** Line 592: It is unclear what is the motivation to study the correlation between human annotations and automatic metrics. Such studies have been conducted by the authors of automated metrics, and it is unclear how it is contextualized in this work. More explanation of insights drawn based on this analysis, which is specifically applicable to this paper, could be useful.
>
> *In this paper, we can observe from human and automatic evaluation that the ROUGE score is very poor at separating the quality of the summaries from BARTHEZ and Open assistant, where the human evaluations show a large difference between them.*
>
> *Comparative studies [2] [3] show that there is no one-size-fits-all metric that can outperform others on all datasets, suggesting the utility of using different metrics for different datasets to evaluate systems. Our experiments show that both reference based and reference-free automatic metrics cannot reliably evaluate GPT-3 summaries. For the above reasons, we want to evaluate this by ourself, and results show that BERTScore correlates well with human judgmnt, and it suggests that BERTSCORE can be a reliable metric for our dataset, for future studies.*
>
> *[2] Manik Bhandari, Pranav Narayan Gour, Atabak Ashfaq, Pengfei Liu, and Graham Neubig. 2020. Re-evaluating Evaluation in Text Summarization. In Proceedings of the 2020 Conference on Empirical Methods in Natural Language Processing (EMNLP), pages 9347–9359, Online. Association for Computational Linguistics.* **Page 6**
>
> *[3] Goyal, Tanya, Junyi Jessy Li, and Greg Durrett. "News summarization and evaluation in the era of gpt-3." arXiv preprint arXiv:2209.12356 (2022).* **Abstract**
>
> ------------------------------------------
>
> Thank you again for the review, we hope to have cleared up the points presented in your reasons to reject

---

### Official Review · Reviewer_P9qY · 2023-08-10

**Soundness:** 4

**Excitement:**

4: Strong: This paper deepens the understanding of some phenomenon or lowers the barriers to an existing research direction.

**Paper Topic And Main Contributions:**

The article presents a new dataset for multi-party dialogue summarization; FREDSum. The dataset consists of French political debates that were manually transcribed. Then, the debates are segmented by dialogue topic (established by the debate moderator) into sub-debates. Those sub-debates are annotated with additional information such as abstractive and extractive summaries, abstractive community annotations, important topics, and evaluation data.

To describe the dataset, the authors present essential sample statistics (number of transcriptions, tokens per transcription, turns per transcription, speakers per transcription, and tokens per abstractive summaries). Also, they carry a study about how models with abstractive summarization capabilities (BARThez, Open Assistant, and ChatGPT) and extractive summarization ones (Longest Greedy, CoreRank Submodular, and BERTExtSum) perform in this new dataset, establishing a baseline. Finally, a human evaluation is conducted to measure the quality of the annotated and generated summaries.

**Questions For The Authors:**

A) Lines 100-105. How does your dataset promote multilingual dialogue summarization techniques? Since the dataset is focused on the French language, it could be to promote language diversity in the research field, but never multilingual promotion.

B) Table 1. Are tokens equal to words? If so, why did you not use "words" instead of tokens? If not, how did you tokenize the text to extract the statistics?

C) Table 1. What are FREDSumPreS and FREDSumPreA? Missing on the caption and/or text.

D) Lines 350-354, 361-364. Could it be the fact of splitting the transcriptions for summarizing with BARThez and Open Assistant, that penalized the quality of the final summary for these models?

E) Lines 544-547. How affects the hallucinations in the score? Is it about the severity of the hallucinations, their amount, or a combination of these two?

F) How is Table 9 created? Is the Pearson correlation calculated with all the generated summaries?




**Reasons To Accept:**

The paper presents a new dataset that is constructed purelly with human annotation data. The dataset focuses on French political debates, with a variety of additional annotations for these debates. Also, baselines are provided as the basis for further research.

**Reasons To Reject:**

Human annotation procedures are not detailed extensively and provoke the following weakness:

- It is unclear whether just one annotator selects the relevant information for a given sub-debate or is consensual by more than one annotator.

- It is unclear how many people participate in the annotation and how many coincide in the annotation and evaluation of the summaries, which could lead to a particular bias. Also, how many of them are part of the authoring of the article.

- It is unclear how the Human annotation scale is designed, what makes a given sample obtain a score between 1 and 5 for a given characteristic (readability, informativity, faithfulness)

**Reproducibility:**

1: Could not reproduce the results here no matter how hard they tried.

**Reviewer Confidence:**

4: Quite sure. I tried to check the important points carefully. It's unlikely, though conceivable, that I missed something that should affect my ratings.

**Typos Grammar Style And Presentation Improvements:**

- 74: Thus, there is a significant

- 110-118: Twitter

- 110-118: Too long sentence; try to split it into parts.

- 142: The debates in FREDSum cover a diverse

- Table 5: It seems to me that there should be 239 words instead of 684 words.

- Table 8: Faithfulness (column head name)

---

> ### Author Rebuttal · Authors · 2023-08-28
>
> We would like to thank you for your detailed and insightful review. We respond point-by-point to your comments below.
>
> ------------------------------------------
>
> **Reasons to accept :**
>
> The paper presents a new dataset that is constructed purely with human annotation data. The dataset focuses on French political debates, with a variety of additional annotations for these debates. Also, baselines are provided as the basis for further research.
>
> *We thank the reviewer for the positive assessment of our work. We are glad that you found our dataset is equipped with a variety of additional annotations. We will address
> the comments and include our response where relevant, especially regarding the annotation procedure, in the next version of the paper.*
>
> ------------------------------------------
>
> **Reasons to Reject :** Human annotation procedures are not detailed extensively and provoke the following weakness: It is unclear whether just one annotator selects the relevant information for a given sub-debate or is consensual by more than one annotator.
>
> *Each annotator selected information and made the summaries independently from other annotators. While it is hard to calculate inter annotator agreement for tasks like extractive summarization, we can however estimate the inter-annotator agreement by calculating the ROUGE score between extractive summaries*
>
> |                              | R1   | R2   | RL   | BERTScore |
> |------------------------------|------|------|------|-----------|
> | Extractive 1 vs Extractive 2 | 0.72 | 0.59 | 0.69 | 0.95      |
>
> *As we can observe from this additional experiment, the agreement between annotators for the creation of extractive summaries is high. Those scores will be added to the appendix of the paper*
>
> ------------------------------------------
>
> **Reasons to Reject :** It is unclear how many people participate in the annotation and how many coincide in the annotation and evaluation of the summaries, which could lead to a particular bias. Also, how many of them are part of the authoring of the article.
>
> *Five annotators overall worked on the annotations available in the paper, three of them being co-authors. Two co-authors created abstractive summaries. A third coauthor contributed to summary evaluation in collaboration with two independent annotators.*
>
> *Of course, while bias is inevitable in political annotations, the overall agreement calculated shows that the annotators mainly agreed on important points. It is important to note that political bias is prevalent in LLMS [1], and the creation of a dataset that has multiple points of view from different summaries can only try to lessen such bias*
>
> *[1] Feng, Shangbin, et al. "From Pretraining Data to Language Models to Downstream Tasks: Tracking the Trails of Political Biases Leading to Unfair NLP Models." arXiv preprint arXiv:2305.08283 (2023).*
>
> ------------------------------------------
>
> **Reasons to Reject :** It is unclear how the Human annotation scale is designed, what makes a given sample obtain a score between 1 and 5 for a given characteristic (readability, informativity, faithfulness)
>
> *Here are the guidelines given to annotators for each of readability, informativity, faithfulness*
>
> *Readability :*
>
> 1. The given summary is incomprehensible
> 2. An overall theme can be gathered from reading the summary, but the sentences are heavy or repetitive
> 3. The summary is readable with little repetition, and the points come across easily
> 4. The summary reads fluently
> 5. The summary reads fluently, and is elegant
>
>
> *Informativity :*
> 1. The summary captures no information of the original debate
> 2. The summary captures the general theme of the original debate (Speakers + subject e.g. Ukraine war, retirement age, immigration)
> 3. The summary captures the general theme of the original debate, as well as the main points made by the different debaters
> 4. The summary captures the general theme of the original debate, the main points made by the different speakers, as well as the responses from the opposite parties
> 5. The summary perfectly captures the nuances of the original debate
>
> *Faithfulness :*
> 1. The summary hallucinates the entirety of the debate
> 2. Major hallucinations, such as mistaking the participants of the debates or the main theme of the debate.
> 3. The summary is faithful enough to be read, but has some minor fabrications, a point made by a speaker is however modified enough to introduce confusion.
> 4. The summary is largely faithful to the original debate, and only includes hallucinations that could be made by a human because of ambivalent wordings in the original transcript
> 5. The summary is entirely faithful
>
> ------------------------------------------
>
> **Questions for the Authors :** How does your dataset promote multilingual dialogue summarization techniques? Since the dataset is focused on the French language, it could be to promote language diversity in the research field, but never multilingual promotion.
>
> *Models now thrive in multi-linguality. LLAMA, GPT-3 and others perform well in multi-lingual settings by training themselves majoritarily on English, with very small subsets of data in different languages. We believe that creating data for complicated tasks such as long dialogue summarization in different languages would help improve the quality of the models trained for those tasks in multi lingual settings*
>
> ------------------------------------------
>
> **Questions for the Authors :** Table 1. Are tokens equal to words? If so, why did you not use "words" instead of tokens? If not, how did you tokenize the text to extract the statistics?
>
> *This is our mistake; we meant words instead of tokens and will correct the problem in the paper*
>
> ------------------------------------------
>
> **Questions for the Authors :** Table 1. What are FREDSumPreS and FREDSumPreA? Missing on the caption and/or text.
>
> *PreS represents the statistics of the Senate subset of the pretraining data, while PreA represents the Assembly*
>
> ------------------------------------------
>
> **Questions for the Authors :** Lines 350-354, 361-364. Could it be the fact of splitting the transcriptions for summarizing with BARThez and Open Assistant, that penalized the quality of the final summary for these models?
>
> *Both Barthez and OpenAssistant score significantly lower than ChatGPT in debates they can fully treat without segmentation; Indeed, the longer the debate, and the larger the amount of cuts, the lower the score is. It is however impossible for both of those systems to summarize entire meetings without the use of segmentation strategies. We report in the rebuttal of Review #3 the scores of each model when no segmentation strategy is applied, and they are only treating the longest string they can get in one pass.*
>
> ------------------------------------------
>
> **Questions for the authors :** Lines 544-547. How affects the hallucinations in the score? Is it about the severity of the hallucinations, their amount, or a combination of these two?
>
> *As shown by the guidelines provided above, the number of hallucinations are taken into account, however, the nature of the hallucinations are also extremely important. We describe the nature of different types of hallucinations in our response to reviewer number two in decreasing order of severity*
>
> ------------------------------------------
>
> **Questions for the authors :** How is Table 9 created? Is the Pearson correlation calculated with all the generated summaries?
>
> *All the generated summaries and all their ratings. Each summary has at least two ratings by human annotators as well as their ratings with respect to the metrics. Each score is calculated by the pearson correlation of all the ratings from all the summaries and their respective scores to each metrics . As an example, consider an abstractive summary A with two human ratings, H1 and H2. To get the Pearson correlation for ROUGE 1, we take the Pearson correlation between the ROUGE 1 score for H1 and the Pearson correlation between the ROUGE 1 score and H2 and average these two scores together. We then do this for all transcripts and take the average of those scores.*
>
> ------------------------------------------
>
> Thank you again for the review, we hope have cleared up the points presented in your reasons to reject

---

### Meta-Review · Area_Chair_ukp1 · 2023-09-12

**Recommendation:** 3

**Metareview:**

The paper introduces a new dataset for multi-party dialogue summarization in French. It consists of transcribed political debates, which are segmented into sub-topics and annotated with abstractive and extractives summaries. The paper reports model performance on this dataset for several existing summarization models and alignment approaches (for aligning summaries with original speech).

The reviewers raise several questions about the annotation process which are not sufficiently explained in the paper, and raise concerns regarding use-policies of the underlying data (the authors address these satisfactorily in their rebuttal).

---

### Decision · Program_Chairs · 2023-10-07

**Decision:**

Accept-Findings

**Comment:**

The paper introduces a new dataset for multi-party dialogue summarization in French. It consists of transcribed political debates, which are segmented into sub-topics and annotated with abstractive and extractives summaries. The paper reports model performance on this dataset for several existing summarization models and alignment approaches (for aligning summaries with original speech).

The reviewers raise several questions about the annotation process which are not sufficiently explained in the paper, and raise concerns regarding use-policies of the underlying data (the authors address these satisfactorily in their rebuttal).